# The auditory brainstem response to natural speech is not affected by selective attention

**Thomas J. Stoll**[1,2], **Nathan D. Vandjelovic**[3], **Melissa J. Polonenko**[4,5], **Nadja R. S. Li**[6], **Adrian K. C. Lee**[6,7], **Ross K. Maddox** [1,2,5,6]*

**1** Kresge Hearing Research Institute, Department of Otolaryngology–Head & Neck Surgery, University of Michigan, Ann Arbor, Michigan, United States of America, **2** Department of Biomedical Engineering, University of Rochester, Rochester, New York, United States of America, **3** Department of Otolaryngology, University of Rochester, Rochester, New York, United States of America, **4** Department of Speech-Language-Hearing Sciences, University of Minnesota, Minneapolis, Minnesota, United States of America, **5** Department of Neuroscience, Del Monte Institute for Neuroscience, University of Rochester, Rochester, New York, United States of America, **6** Institute for Learning & Brain Sciences (I-LABS), University of Washington, Seattle, Washington, United States of America, **7** Department of Speech & Hearing Sciences, University of Washington, Seattle, Washington, United States of America

* rkmaddox@umich.edu

## Abstract

The ability to pick out and attend to one sound in a noisy mixture underpins successful communication in many natural scenarios. Selective attention has been shown to drastically alter sound encoding in the cortex and has been hypothesized to modulate subcortical processing via an extensive efferent network. The body of work investigating this possibility in humans has not resulted in a clear consensus, possibly owing to limitations in the available methodologies. Here, we used new experimental tools that allowed distinct neural responses from across the auditory pathway to be simultaneously acquired from human listeners attending to the sound of one person talking while ignoring a second. Our series of experiments provide convergent evidence that selective attention does not influence sound encoding in the auditory periphery or brainstem, with an effect first appearing in the cortex. Humans' ability to focus their attention on a single sound even in the presence of many others is as remarkable as it is essential. These findings add needed clarity regarding the mechanisms that make this feat possible.

## Introduction

Attention is fundamental to parsing and processing the sensory world. For the auditory system, selectively attending to one sound in a mixture is a complex process that involves extraction of features and segregation and selection of auditory objects [1]. The effects of selective attention on neural encoding are clearly visible in the auditory cortex, with attended sounds represented robustly while ignored sounds are suppressed [2,3]. There are extensive efferent projections from the cortex to subcortical

**Data availability statement:** All code, EEG recordings, and stimulus files are available in BIDS format on the OpenNeuro database (https://openneuro.org/datasets/ds006434, DOI: https://doi.org/10.18112/openneuro. ds006434.v1.2.0).

**Funding:** This work was supported by the National Institute on Deafness and Communication Disorders (nidcd.nih.gov) grants R00DC014288 (awarded to RKM) and R01DC013260 (awarded to AKCL), and National Science Foundation (nsf.gov) grant 2142612/2448814 (awarded to RKM). The funders did not play any role in the study design, data collection and analysis, decision to publish, or preparation of the manuscript.

**Competing interests:** The authors have declared that no competing interests exist.

**Abbreviations:** ABR, auditory brainstem response; ANM, auditory nerve model; BF, Bayes factor; CAP, compound action potential; FDR, false discovery rate; FFT, fast Fourier transform; FFR, frequency-following response; FIR, finite impulse response; ICA, independent component analysis; OAE, otoacoustic emission; OCB, olivocochlear bundle; TRF, temporal response function.

auditory areas [4,5]. These projections could modulate subcortical processing in numerous ways, with some suggesting a top-down "filtering" of early activity in service of selective auditory attention [4].

The evidence for a subcortical effect of attention in human listeners is best described as mixed. Some studies have reported an effect [6–22], though results are inconsistent across and sometimes within studies, while others have found no effect [23–33]. Prior work in this area, however, has faced limitations. Most studies have been restricted to using simple, artificial sounds to evoke a measurable subcortical response, making it difficult to design an engaging selective attention task. Several studies have also used the electroencephalography (EEG) frequency-following response (FFR), whose generators are now known to include the cortex in addition to a mixture of subcortical areas [34–36]. We have recently developed a set of stimulus and analysis techniques using natural speech to obtain neural responses [37–39] whose distinct components can be clearly tied to subcortical or cortical areas. Applying these tools to the problem of selective attention should provide a clear determination of where in the auditory pathway effects of selective attention can first be seen.

Here, we performed a series of experiments in which subjects were presented with two simultaneous audiobooks and instructed to selectively attend one of them while we recorded brain activity at the scalp using EEG. Using our recently developed tools, along with a new electrode that rests on the eardrum [40], we simultaneously acquired the compound action potential (CAP) of the auditory nerve, the auditory brainstem response (ABR), and later cortical responses. The first two experiments showed no subcortical effect of attention in either diotic listening (i.e., without binaural cues) or dichotic listening (one narrator in each ear), while replicating the large cortical effects seen in previous studies [2,3]. Our finding of no subcortical attention effect contradicted a set of recent studies that also used natural speech [6,7,11]. To reconcile this difference, we conducted a third experiment in which subjects listened passively to the stimuli from those studies. We found the previously reported effect to be spuriously driven by acoustic differences between the stimuli, rather than attention.

## Results

### Selective attention impacts cortical but not subcortical encoding of speech

We presented subjects simultaneously with one audiobook read by a male narrator [41] and another read by a female narrator [42] and asked them to attend to one or the other story in each trial. The stimuli were presented diotically by adding the audio together and presenting the same audio in both ears. The stories were ordered such that subjects could follow along, with each stimulus played twice (once attended, once ignored), so they would not miss sections of the story (see Table 1). At the end of each trial, subjects were asked two comprehension questions to keep them engaged in the task.

As the subjects listened to the stories, we used EEG to record both cortical and subcortical responses. In 12 of the 28 subjects, we additionally obtained auditory nerve activity using an eardrum electrode. Since subcortical responses to speech cannot be determined using standard averaging methods, we used the temporal

**Table 1. Stimulus presentation structure.**

| Trial number | 1 | 2 | 3 | 4 | 5 | 6 | 7 | 8 | 9 | 10 | … | 116 | 117 | 118 | 119 | 120 |
|---|---|---|---|---|---|---|---|---|---|---|---|---|---|---|---|---|
| Story segment | 1 | 2 | 1 | 2 | 3 | 4 | 3 | 4 | 5 | 6 | … | 58 | 59 | 60 | 59 | 60 |
| Attended talker | F | F | M | M | M | M | F | F | F | F | … | M | M | M | F | F |

response function (TRF) framework, adapted for the speed of the subcortical auditory system [37–39]. The auditory nerve, brainstem, and cortical responses were separately but simultaneously obtained by considering different electrode-latency combinations (CAP ~3 ms using the eardrum electrode referenced to the ipsilateral earlobe; ABR before ~15 ms using FCz referenced to the earlobes; cortical responses from ~15 ms and beyond using 32 channels referenced to the mastoids). The individual peaks of the ABR can be tied to specific subcortical areas, with the largest peak indexing the rostral brainstem [43], labeled as wave V.

Figs 1a and 1b show the grand average subcortical responses to the attended (red) and unattended (black) speech stimuli for the CAP and ABR, respectively (individual subject responses are in Supplemental S1 and S2 Figs). No significant differences were observed for either the CAP or ABR (paired two-tailed *t* tests from 0–7 to 0–15 ms for the CAP and ABR, respectively; $p > 0.05$ at all time points, false discovery rate (FDR) corrected). To quantify if this result reflected an absence of evidence of an effect or evidence of absence, we determined the peaks of the CAP and wave V for each subject (Fig 1c and d, respectively) and performed *t* tests and calculated Bayes factors (BFs) using the response size and latency differences between conditions (Table 2). None of the *t* tests showed a significant effect of attention, consistent with the previous per-time point analysis. BF compares the relative likelihood of the null hypothesis to the alternative [44]. BFs for the CAP and wave V latency were too small to determine evidence of absence, likely due to the smaller number of subjects fit with the eardrum electrode. The BF for ABR wave V peak size, however, provided evidence *against* the hypothesis that wave V's size was larger for attended speech than unattended ($BF_{0+} = 5.7$). Although the BF for wave V peak latency could not provide evidence against an attention effect, we note that the difference in wave V latencies between conditions was within one sample (0.1 ms) for all subjects, with a difference of exactly 0.0 ms in 14 of 28 subjects. Of the remaining half of the subjects, 10 of the responses were one sample later in the unattended condition than the attended, while four were one sample earlier. Thus, even if there were an effect on latency, it would be so small that it was essentially unmeasurable at a sampling rate of 10 kHz. In addition to the grand averages and summary statistics, visual examination of the 12 individual-subject CAPs and 28 individual-subject ABRs shows that there are no discernible differences between conditions for any subject (S1 and S2 Figs).

Cortical responses were calculated in the same manner as the ABR, but with the recordings of 32 electrodes across the scalp and after applying a low-pass filter at 20 Hz (see Methods for all filtering details). Figs 1e and 1f show the cortical responses for the attended and unattended conditions, respectively. Significant differences were observed across a wide time range from 38 to 271 ms (paired, two-tailed, nonparametric spatiotemporal clustering permutation tests, $p < 0.05$ for at least one electrode in the shaded region; Fig 1g). Consistent with prior literature, we observed an enhanced negative-going potential around 100 ms for the attended condition, and activity was strongest in the frontocentral scalp electrodes. This clear effect confirms that subjects were engaged and attentive. Therefore, in this selective auditory attention task, we find that attention first influences neural encoding of sounds in the cortex.

## Subcortical selective attention effects are still absent when stimuli are presented to separate ears

In the previous experiment, sounds were presented identically to each ear, requiring any putative attentional mechanism to first segregate overlapping speech streams before one's representation may be enhanced. If the sounds were different in each ear, a much simpler scheme could be employed wherein the better ear was favored. Thus, we performed another experiment in which the audiobooks were presented separately to each ear. Subjects were again instructed to attend to a

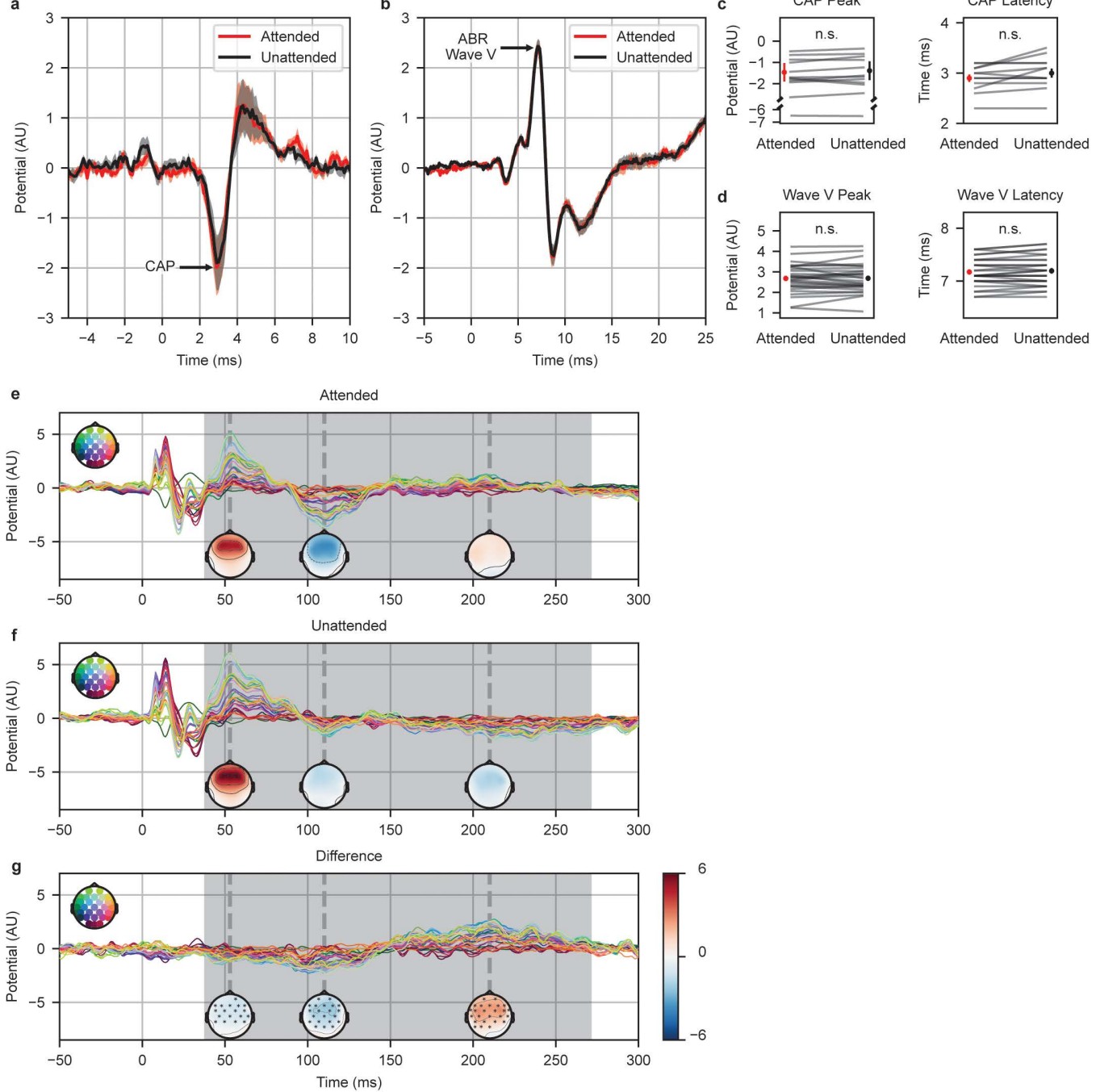

**Fig 1. Subcortical sound encoding is not affected by selective attention, while cortical sound encoding is. a**, The CAP to the attended (red) and unattended (black) speech for the 12 subjects with the eardrum electrode. **b**, The ABR for all 28 subjects when attended (red) or unattended (black). No significant differences were observed between conditions for either the CAP (paired two-tailed $t$ tests in the 0–7 ms time window, FDR corrected, $p > 0.05$ at all points) or ABR (paired two-tailed $t$ tests in the 0–15 ms time window, FDR corrected, $p > 0.05$ at all points). Shaded regions represent SEM. **c**, CAP peak potentials and latencies. There is a break in the Potential axis due to one subject with a substantially larger CAP. **d**, ABR wave V peak potentials and latencies. The peak potential sizes and latencies were not significantly different for either the CAP or ABR Wave V (peak potentials: paired one-tailed $t$ tests, FDR corrected, $p > 0.05$; peak latencies: paired two-tailed $t$ tests, FDR corrected, $p > 0.05$). The data underlying c and d can be found in https://doi.org/10.18112/openneuro.ds006434.v1.2.0 at code/Results/exp1Diotic/mags+lats.csv. Cortical responses to attended and unattended stimuli are shown in **e** and **f**. Each electrode is represented by a different color trace (key in upper left). Significant differences were observed in the 38–271 ms

region (shaded on the time-series plots), as determined through paired, two-tailed spatiotemporal clustering methods ($p < 0.05$ for at least one electrode in the interval). Scalp topographies are shown for selected time points of 53, 110, and 210 ms, denoted by a vertical dashed line on the time series plots. **g**, The difference waveform and scalp topographies at the selected time points. Asterisks indicate the electrodes that were significantly different across conditions at that time.

**Table 2. Statistical analysis of picked peaks.**

| | Test direction | t | DOF | $P_{FDR}$ | $BF_{01}$ |
|---|---|---|---|---|---|
| CAP peak potential (size) | Lower | −1.29 | 11 | 0.15 | 1.0 |
| CAP peak latency | Two-tailed | −2.03 | 11 | 0.15 | 0.8 |
| Wave V peak potential (size) | Upper | −0.19 | 27 | 0.57 | **5.7**[†] |
| Wave V peak latency | Two-tailed | −1.65 | 27 | 0.15 | 1.5 |

Statistical analysis of CAP and ABR wave V peak sizes and latencies. No significant differences are found through t tests ($p > 0.05$ for all comparisons, FDR corrected). The Bayes factors (BF) are inconclusive for the CAP measures and wave V latency but provide moderate evidence ($3 \leq BF < 10$, indicated by †) against an effect in wave V size.

specific audiobook on each trial, now with an added directional cue. The same trial order was used so that subjects could follow the stories. No eardrum electrode was used in this experiment, so we report only on the ABR. Due to subtle differences in the stimuli from the previous experiment (see Methods), more stimulus artifact was present in these recordings, and response SNR is lower. Stimulus artifact was minimized using methods previously described [45], wherein an estimate of the continuous stimulus artifact is subtracted from the EEG recording prior to response calculation.

Comparing the ABRs for the attended and unattended conditions (Fig 2), we saw no significant differences between the responses at any time point ($p > 0.05$ at all points in the 0–15 ms range, FDR corrected). We again determined ABR Wave V peak size and latency and compared them between the responses to attended and unattended speech. The t tests indicated no significant effect of attention, consistent with the per-time point analysis. The BF for the hypothesis that ABR wave V size was larger to attended speech than unattended indicated no evidence for or against ($BF_{0+} = 1.6$). However, the BF examining wave V latency provided evidence against the hypothesis of a difference in latency between the responses to attended and unattended speech ($BF_{01} = 4.2$). As above, selective attention showed clear effects in the later cortical responses (S3 Fig). Taken together with the findings of the previous experiment, we find no evidence that selective auditory attention influences subcortical sound encoding, with BF providing evidence against such an effect in several cases.

## Explaining recent contradictory results

**Alternative analyses applied to our data do not reveal an attention effect.** Forte and colleagues [7] and the two following studies [6,11] used a similar competing speech paradigm to investigate subcortical selective attention and reported an effect. To examine the discrepancy between our results and theirs, we first tested if the complex cross-correlation metric used in those studies (which provides a single broad peak at ~9 ms) would show a different result than our analyses if applied to our data. We calculated those responses by bandpass filtering our stimuli and EEG data from 100 to 300 Hz, computing the cross-correlation, and extracting the Hilbert envelope. We then determined the peak value of each response in the 0–20 ms time window for both attention conditions, shown in Fig 3 (individual subject waveforms shown in S4 Fig), and confirmed that no responses were maximal on the edge of the selected time window. The response to the attended stimuli was maximal at a mean latency of 7.9 ± 0.3 ms, and the response to the unattended stimuli was maximal at a mean latency of 8.0 ± 0.4 ms, indicating the calculated responses likely originate from subcortical neural generators. Sixteen subjects showed a larger response to the attended stimuli versus 12 to the unattended stimuli, a result which is not significantly different from chance (binomial test, $p = 0.57$). We computed the average ratio of response

PLOS Biology

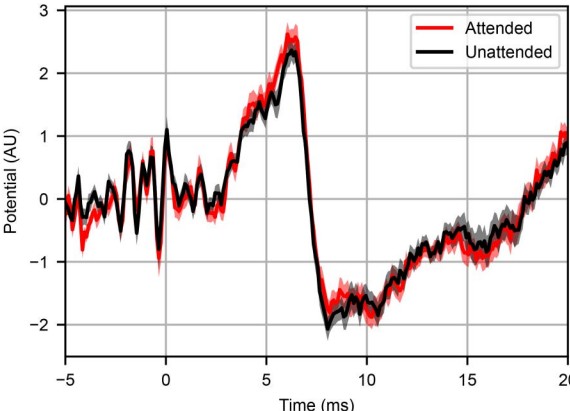

**Fig 2. Subcortical sound encoding is unaffected by selective attention in a dichotic listening task.** The ABR to dichotic speech when attended (red) or unattended (black) for 24 subjects shows no significant differences between conditions (paired two-tailed $t$ tests in the 0–15 ms time window, FDR corrected, $p > 0.05$ at all points). Shaded areas represent SEM.

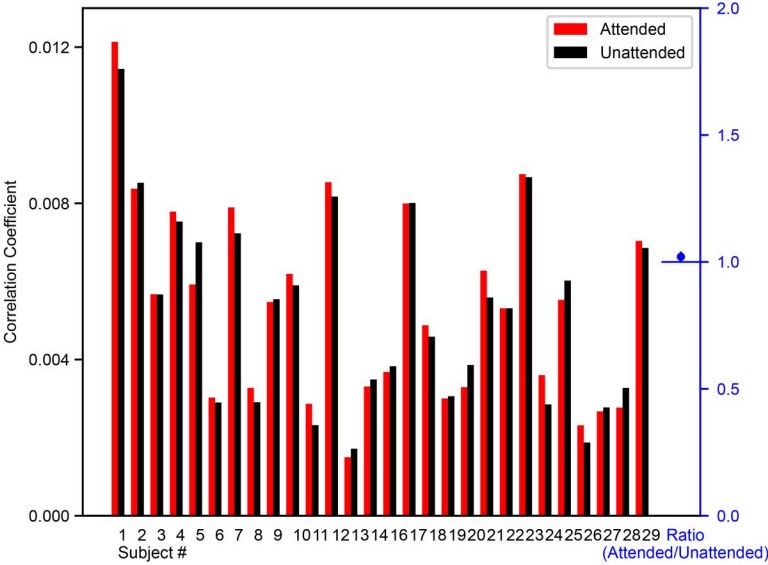

**Fig 3. Complex cross-correlation analysis shows no selective attention effects in subcortical responses.** The maximum value of the response for the attended (red) and unattended (black) conditions is shown for all subjects ($n = 28$, data from the first experiment). The mean ratio (attended ÷ unattended; shown in blue at right) was not significantly greater than unity (one-tailed upper $t$ test, $p > 0.05$). Error bars represent SEM.

sizes (attended ÷ unattended) to be 1.02 ± 0.02, which was not significantly greater than unity (paired one-tailed upper $t$ test $p = 0.23$). Therefore, applying the metric from Forte and colleagues to our dataset again provides no evidence that selective auditory attention affects subcortical auditory activity.

**Acoustic differences between stimuli explain the reported effect.** A second way in which the prior studies differ from the present one is in the stimulus design. Forte and colleagues used different stories for the attended and unattended conditions (one attended and one unattended story spoken by each of the two narrators). This design allows each stimulus to be presented only one time during the experiment (as opposed to twice—once attended and once

unattended), but it introduces uncontrolled acoustic differences in the stimuli used across conditions that could drive the observed effect, even though the four different stories were still spoken by the same two narrators. We conducted a passive listening experiment to test this possibility using the same stimuli and analysis methods described in Forte and colleagues, but providing no task instructions, instead allowing subjects to watch captioned silent videos of their choice or read quietly. Here, we term the stimuli used for the attended condition in the Forte and colleagues paper as "target" and the unattended condition as "distractor" to distinguish between the previous sections of the paper where stimuli were truly attended or unattended.

Twenty-two out of 28 comparisons showed a larger response to the "target" than the "distractor" (14 subjects × male and female narrator), which is significantly greater than chance (binomial test, $p = 0.004$). Response amplitudes are shown in Fig 4 and individual subject responses in Supplemental S5 Fig. Additionally, the ratio of the "target" to the "distractor"

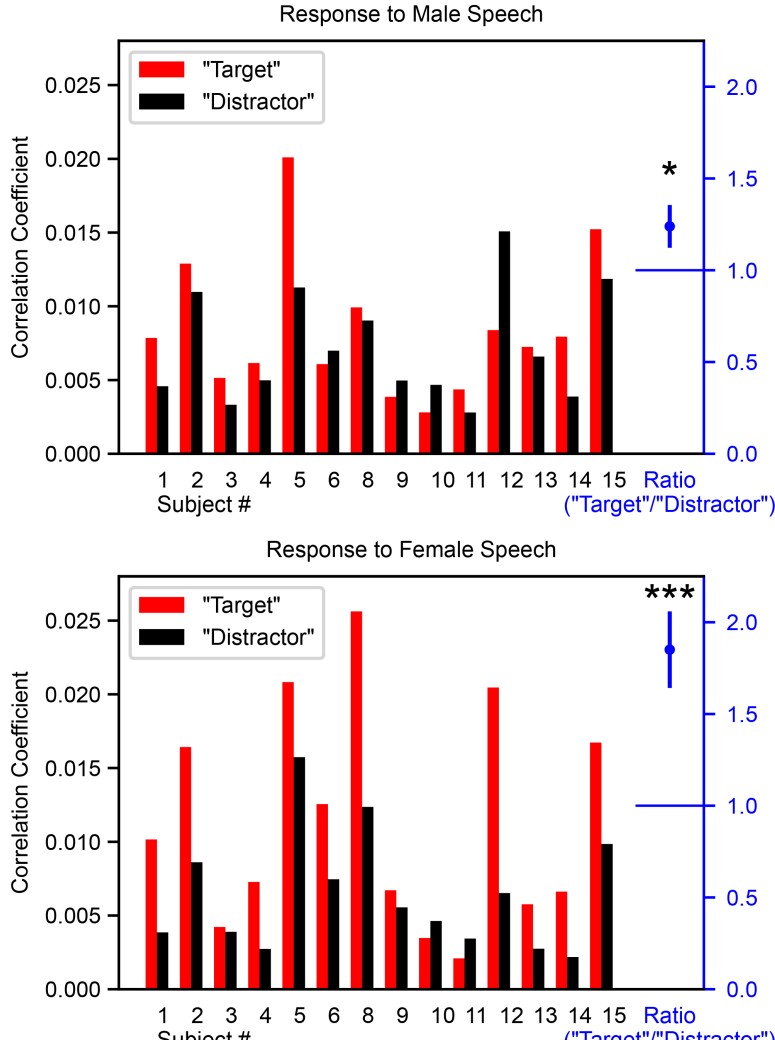

**Fig 4. Stimulus-specific acoustics result in response size differences during a passive listening experiment.** Bar plots show the peak size of responses determined through the complex cross-correlation metric as each subject passively listened to the "target" and "distractor" stimuli from Forte and colleagues. The mean ratio ("target" ÷ "distractor"; shown in blue at right) was greater than unity for both the male (ratio = 1.24 ± 0.12, paired one-tailed upper $t$ test, $p = 0.034$) and female speech (ratio = 1.86 ± 0.21, paired one-tailed upper $t$ test, $p = 0.0009$). Error bars represent SEM.

maximum response size was significantly greater than unity for both the male speech (ratio = 1.24 ± 0.12, $p = 0.034$, paired one-tailed upper $t$ test) and the female speech (ratio = 1.86 ± 0.21, $p = 0.0009$, paired one-tailed upper $t$ test). This indicates the "target" stimuli used as the attended condition in Forte and colleagues provide significantly larger responses than the "distractor" stimulus even when subjects are only passively exposed to the stimuli, providing an explanation for the differences observed.

## Discussion

Leveraging several recent methodological advances, we investigated the neural effect of selectively attending to one speech stream in an ecologically relevant "cocktail party" task. We simultaneously measured responses spanning the human auditory system, from the periphery through the cortex. We found no effect of attention in the auditory nerve and no effect in the auditory brainstem (with Bayesian analysis providing evidence *against* attention effects in the size of responses there). We did find a strong cortical effect of attention, replicating many prior studies and confirming the validity of our approach.

The goal of this study was to determine where in the auditory system selective attention first impacts auditory encoding. Standard EEG electrode montages afford clear responses from later subcortical stages (e.g., ABR wave V), but the ABR component originating from the auditory nerve (wave I) cannot be reliably measured from speech in every subject and is inconveniently small when present (see Fig 1). The combination of a recently developed eardrum electrode [40] and our stimulus [37] and analysis [38,39] paradigms allowed us to measure the CAP—an auditory nerve response with much higher SNR than wave I—in response to speech. Neither the size nor the latency of the CAP to attended speech differed significantly from those to unattended speech (see Fig 1a and 1c). We also tested brainstem encoding via ABR wave V, again finding no effect of attention on response size or latency (see Figs 1b, 1d, and 2). In addition to finding no significant effects, BF indicated moderately strong evidence against an effect of attention in the size of brainstem-generated wave V. There was no strong evidence either way in the auditory nerve, possibly due to the smaller number of subjects with the eardrum electrode applied. Still, an attention effect in the auditory nerve seems unlikely, as it would have to vanish in the brainstem only to reemerge in the cortex several synapses later. The simpler explanation of these results is that attention's effect arises in the cortex de novo—leaving the periphery and brainstem to encode and extract features of incoming sounds, regardless of which is attended.

Separating overlapping speech signals so that one can be selectively enhanced is nontrivial, especially when both are equally present in both ears, as above. For this selective enhancement to be present subcortically, the modulating efferent feedback would need to be dynamic and well synchronized with the attended stream. If the attended and unattended signals were spatially separated, then each one would appear at a higher sound level than the other in one of the two ears. In that case, an effective and simple strategy would be to rely on the ear that favored the attended signal. We tested for such a mechanism using the extreme case of dichotic presentation—i.e., presenting each ear with one isolated speech stream. This second experiment again revealed no significant attention effect in the brainstem. Bayesian analysis indicated evidence against an effect on wave V latency, with no strong evidence either way for an effect on wave V size (which we attribute to lower SNR in that experiment, due to reasons discussed in the Methods). The combination of this second experiment's outcome with those of the first indicates that selective attention does not enhance subcortical speech encoding by any mechanism.

Efferent pathways are pervasive throughout the auditory system and influence subcortical sound encoding in several ways [46–48], with additional functions surely still to be discovered. At the lowest level, the periphery receives projections from the superior olivary complex via the olivocochlear bundle (OCB). The OCB has been has been shown to play a role in gain control [49] and protection against high sound levels [50], with several other possible functions suggested as well [46]. Rapid and frequency-specific filtering would be needed to selectively enhance the AN response to one sound in a mixture, but prior research has suggested that effects of the medial OCB may be both too slow to keep up with speech's dynamic structure and too broadband to be selective [49]. Consistent with this, we found no effect of selective attention in the auditory nerve (assessed with the CAP) in our experiment.

A selective attention effect could arise later in the subcortical pathway, such as the inferior colliculus, which is the recipient of extensive efferent corticocollicular projections [51]. Like the olivocochlear bundle, several roles for these efferents have been suggested [52,47], with selective attention among them. Yet here as in the periphery, we find evidence against selective attention effects in the inferior colliculus or other midbrain nuclei, indicated by perfectly overlapping ABR waveforms and no wave V differences. Therefore, our results indicate that efferents with subcortical targets are not selectively enhancing the encoding of individual speech streams in a mixture, whether they are superimposed or presented to separate ears. We note, however, that these findings should not be construed as indicating that subcortical efferents don't affect subcortical stimulus encoding in other important ways. Additionally, our results also do not indicate that no efferent projections are involved in selective attention—it seems a near certainty that top-down attention relies on at least intracortical efferent projections, but that is beyond the scope of what this study was designed to investigate.

## Reconciling our findings with prior studies of subcortical attention effects

This study's focus was specifically auditory selective attention (i.e., attending to one of multiple simultaneous sounds). Subcortical attention effects have been studied many times, in many ways, over many decades, with varying findings. In order to reconcile the present results with prior studies, we divide the previous studies into two groups. The first group comprises studies investigating attention but not specifically auditory selective attention. These studies have investigated intermodal attention (e.g., focusing on either a visual or auditory stimulus) [14–16,18–21,29–33] or active versus passive listening [12,13,53]. Some of these studies have reported effects [12–16,18,19,53], some have not [29–33], and some have had differing results across conditions within the same study [22]. The mechanisms tested in those studies are different than what would be required for auditory selective attention. As such, our results can neither conflict with nor confirm their findings. The second, smaller group of studies is focused on the same attention effects as the present one. We discuss below their mixed results and how our findings bear on the debate, especially with regards to a specific set of recent studies similar to this one [6,7,11], but with a methodological flaw leading to spuriously reported attention effects, as demonstrated by our third experiment.

Otoacoustic emissions (OAEs) allow researchers to probe the function of the outer hair cells of the cochlear epithelium [43]. Outer hair cells receive efferent projections from central areas [43], offering a possible pathway for attentional modulation. Giard and colleagues [9] reported an effect where OAE power was modulated by attention in a band around the stimulus frequency. In their study, 12 subjects performed a dichotic task in which they counted tonebursts presented to one ear while ignoring those in the other. They note that the effect was not visible in the time-domain OAE waveforms, and data shown for selected subjects indicate highly idiosyncratic OAE spectra. Further, their conclusions are based on 12 separate statistical comparisons with no mention made of controlling for multiple comparisons. Michie and colleagues [27] published a study 2 years later comprising six related experiments, including one designed to replicate Giard and colleagues [9]. They concluded that their results, including those of the replication experiment, "do not support active cochlear involvement in selective attention." Similarly, while additional studies [10,54] have reported attention effects on OAEs, the direction of the effect has been inconsistent, and the effect size small. OAE recordings can be contaminated by physiological noise [55], and a more recent, larger study found no effect, and suggested that the previously reported attention effects could be due to noise-driven false positives in small study samples [23].

Several studies have used the time-locked response to short, periodic sounds such as vowels to study subcortical attention effects (often referred to as the FFR). Galbraith and colleagues have reported mixed results across several studies [8,56–58], with one study finding no effect of attention [56] and others finding a significant interaction of selective attention with other experimental factors [8,57,58]. Lehmann and Schönwiesner [59] reported idiosyncratic attentional modulation of individual subjects' FFRs, with the direction of the effect varying from subject to subject. Following that study, Varghese and colleagues [25] found no effect of attention on the FFR. Their discussion of the previous findings noted issues in the work of Galbraith and colleagues, concluding that not accounting for the spectral noise floor and poor

SNR may underpin their conclusions. Varghese and colleagues also described statistical errors in Lehmann and Schönwiesner, concluding that their data do not in fact support a subcortical attention effect. Shortly after the study by Varghese and colleagues, seminal work from Coffey and colleagues [34–36] demonstrated that the FFR, previously thought to originate from a mixture of purely subcortical regions, also included a substantial cortical contribution for frequencies up to and including the typical fundamental frequency of speech—the same frequencies commonly examined in FFR studies. With this knowledge, Holmes and colleagues [24] found an effect of attention in the FFR at the fundamental frequency but not the higher harmonics, consistent with attentional modulations of the FFR being driven by cortical and not subcortical generators.

We identified two studies that examined subcortical selective auditory attention using fMRI, which trades the high temporal resolution of EEG for spatial specificity. Rinne and colleagues [60] presented subjects with dichotic stimuli and instructed them to attend to sounds in one ear and press a button to report the direction of randomly timed pitch changes in the attended stimulus. A global voxel-wise analysis using a general linear model revealed an effect of attention in the auditory cortex but not the inferior colliculus. An additional analysis, involving temporal interpolation and smoothing, was then used to reconstruct the BOLD time course. In the ear and at the time point at which the difference in activations between attention conditions reached its maximum, that difference was miniscule at 0.07%. Inspecting the individual subject time courses in their supplemental data reveals a high degree of noise in each subject's responses, along with substantial variability across subjects, including in the direction and timing of the differences. To our knowledge, no studies have replicated these findings. A later study [61] tested frequency-specific attention in diotically presented stimuli. That study comprised two individual experiments, the first testing auditory cortex and IC together and the second aimed specifically at IC. Neither of them showed a significant effect of selective attention.

We are aware of only one prior study [26] that measured evoked potentials from the time of a transient stimulus through later latencies, including ABRs of high quality. Unlike the FFR, the subcortical components of the transient-evoked response can be readily identified by their earlier latencies. In that study, attention effects were found to begin in the middle latency response components (~20–50 ms), which originate from primary cortical regions [43]—no attention effects were observed in the shorter latency subcortical components. This study, while compelling, was limited to using short, artificial sounds that did not overlap in time so that the ABR could be calculated by averaging many stimulus repetitions. The difficulty of designing a paradigm that engages listeners' natural attentional mechanisms, while using only the transient stimuli needed to evoke the canonical ABR, may explain the absence of similar follow-up studies in the several decades since this one was published.

**Addressing incongruity with recent studies also using natural speech.** Several recently published papers have established and refined methods for studying the subcortical response to speech by using the TRF [37–39,45,62–66] or cross-correlation methods [6,7,11]. In addition to the present study, three recent papers from another research group have used these advancements to study selective attention to simultaneously presented audiobooks [6,7,11]. Contrary to our findings, those studies all reported a subcortical effect of attention. To explain our conflicting findings, we tested two key differences in the ways our studies were conducted.

The first difference was in how subcortical responses were derived. The prior papers computed the responses as the cross-correlation of the fundamental waveform of each speech stimulus with the corresponding EEG. The result was a response with a single broad peak around 9 ms, which was confirmed to be subcortical through modeling [67]. We applied that analysis to the data we collected here in our first experiment, but no effect of attention was revealed (Fig 3). Thus, differences in analysis do not offer an explanation.

The second difference was in how specific stimuli were assigned to experimental conditions. In our study, all stimuli were presented once while attended and once unattended, ensuring counterbalanced comparisons across attention conditions. The prior studies employed four different stories (two read by a female narrator and two by a male), assigning one story to each of the experiment's four conditions (male-attended, male-unattended, female-attended, female-unattended).

This design carries the possibility that differences in stimulus acoustics could underpin the putative attention effects, which we tested in our third experiment. We presented the same stimuli and used the same analysis as Forte and colleagues [7], but importantly did not instruct subjects to attend any of the stimuli (instead they watched a subtitled movie or read quietly, ignoring the audio altogether). We found that the stories that were to be attended in the original experiments produced a larger response than those that would have been unattended (Fig 4). From this, we conclude that the attentional modulations reported in the previous studies were spurious—an artifact of stimulus acoustics. This is consistent with recent work showing that stories with similar acoustics can produce responses of varying size, for reasons that are yet to be understood (see Extended Data Figure 3-1 in Polonenko and Maddox [66]).

It is reasonable to wonder if subjects could have selectively attended the "target" stories in our experiment despite instructions to ignore the stimuli, but the realities of the experiment make that a near impossibility. First, the story segments were presented in a randomized order so that subjects could not follow their plots. Second, each segment, when it came up in the order, was repeated four times in a row. Even if one story segment grabbed a subject's attention the first time, it would be unlikely to do so on the second, third, and fourth presentations. Third, subjects were not given any information about what the stories would be, or even that there were two different stories from each narrator being presented. With the segments presented out of order, it would be hard for them to even ascertain that fact, and they would have had to determine at the beginning of each segment which stories were being presented based on content. Fourth, even if subjects did sometimes listen to the speech, the large majority of subjects would have had to agree on which two stories were worth attending, despite all the above, to see the statistically significant results we found. Finally, subjects brought in their own book to read or chose videos from the internet to watch—they would not (perhaps even could not) have chosen something less engaging than these experimental stimuli. The prior points establish why attention could not have driven the results of our third experiment, under the assumption that selective attention can influence subcortical responses at all. It bears repeating, though, the principal finding of our first and second experiments, where stimuli were counterbalanced: there is no subcortical attention effect, even when using the same analyses as in Forte and colleagues [7] (Fig 3).

In addition to the reported subcortical attentional modulation, Saiz-Alía and colleagues [11] also observed a correlation between the size of that effect and speech-in-noise understanding across individual subjects. Due to the correlation's marginal significance, they recommended that it should be interpreted with caution. Given our findings and some other aspects of their study, we now consider that correlation's significance to be the result of a type I error. We note that the correlation was only significant for the male narrator but not the female (with the female correlation going in the opposite direction). Twenty-one correlations across several metrics (most not discussed here) were tested in that paper and multiple comparisons were corrected by limiting the false discovery rate (FDR) to 10%, a value suited for an exploratory study. Correcting for a more conventional FDR of 5% [68], the correlation no longer meets the threshold for significance.

## Caveats and conclusions

Despite the results of the present study, it is still possible that selective attention modulates subcortical responses in a way that we cannot yet measure. EEG requires thousands of neurons to respond synchronously with some degree of spatial alignment for the electrical activity to be observable on the scalp [43,69]. It is conceivable that some subcortical nuclei or types of neurons may be modulated by attention, but are too few in number or lack the requisite dendritic alignment to be measurable [70]. It is tempting to consider modifying aspects of the experimental design, such as task difficulty, stimulus level, etc., so that an effect might be observed in a new experiment. The existing work in this area, however, has run the experimental gamut and in aggregate does not provide compelling evidence of an effect (see prior discussion). It is also worth considering that "unsuccessful" pilot studies are a universal part of the scientific pursuit, and completed studies with null results are more likely to be rejected by journals—if they are even submitted [71].

In this study, we used a naturally engaging speech listening task and new experimental tools that allowed us to measure neural responses with unambiguous origins spanning the auditory pathway. We ran multiple experiments

to determine if selective attention's effects arise subcortically: the first two experiments testing different hypothetical efferent mechanisms, and the third demonstrating the likely spurious origins of a set of previous studies' findings. We therefore conclude that there is no evidence that selective auditory attention modulates sound encoding below the auditory cortex.

## Methods

### Experiment 1: Diotic stimuli

**Subjects.** We recruited 28 subjects (9 male, 19 female) for the diotic experiment with a mean age and standard deviation of 24 ± 4.8 years (range 18–38). Fourteen of these subjects (4 male, 10 female) had the eardrum electrode in place during the experiment and a mean age and standard deviation of 24 ± 5.5 years (range 19–38). Two subjects were excluded from the CAP analysis due to poor signal from the eardrum electrode resulting in no visible response. All subjects had normal hearing (defined as audiometric threshold ≤20 dB HL at octave frequencies from 500 to 8,000 Hz) which was confirmed by pure-tone audiometry. Subjects reported English as their primary language and no neurological disorders or abnormalities. Written informed consent was obtained from all subjects prior to the experiment and subjects were compensated at an hourly rate. Protocols were approved by the University of Rochester Research Subjects Review Board (No. 1227). All procedures were in accordance with the Declaration of Helsinki.

**Experimental design.** Subjects were seated in a comfortable chair in a sound-isolated and electrically shielded room (IAC, North Aurora, IL, USA) and prepared for EEG recording. The experiment was controlled using custom Python scripts and open-source software [72] to present the stimuli and questions. Sounds were presented using ER-2 insert earphones (ER-2, Etymotic Research, Elk Grove, IL, USA). At the beginning of each trial, subjects were instructed to attend to either the male or female speaker. To prevent a subject becoming distracted and forgetting which narrator should be attended in the middle of the trial, a fixation dot was displayed which was blue or magenta (luminance matched) to indicate the subject should attend the male or female narrator, respectively. Subjects pressed a key to start each trial so that they could take breaks between trials whenever they felt it necessary. At the end of each trial, the subject was asked two multiple-choice comprehension questions, and the subject was given feedback whether their answers were correct. Every 12 trials, subjects were given an update on their progress and overall score on the comprehension questions and instructed to take a longer break. Control analyses showed that neither the experiment duration nor the stimulus counterbalancing affected the results (S6 Fig).

**Stimuli.** The stimuli used were constructed in a previous experiment [37] using two audiobooks (male speech: The Alchemyst [41], read by Denis O'Hare, female speech: A Wrinkle in Time [42], read by Hope Davis). The audiobooks were upsampled to 48 kHz and silent pauses were truncated to 0.5 s. The stories were then split into 64 s intervals, with 1 s cosine fade-in/fade-out. The last 4 s of a trial were repeated as the first 4 s of the next trial, so that subjects could follow along more easily. Over 120 trials, a total of 128 min of stimuli were presented. Our prior work has shown narrators with lower fundamental frequencies produce larger responses [62]. Here, the fundamental frequency of the female story was artificially lowered halfway to that of the male story to improve SNR of the responses to female speech. This change did not affect the perceived gender of the talker. For each story, the glottal pulse times were extracted using PRAAT [73]. The acoustic waveforms were then resynthesized using cosines with the phases of each harmonic aligned at each glottal pulse time. By aligning the phases of the cosines and matching the spectrogram of the reconstructed stimulus with that of the original stimulus, the resulting acoustic waveform has sharp peaks when glottal pulse occurs while the spectrotemporal properties of the original stimulus are preserved, thus generating a stimulus which sounds nearly identical to the original speech (example audio available online [37]). Each trial was normalized to have the same root mean square value, and the male and female speech were summed together to be presented diotically at 68 dB SPL (i.e., each stimulus at 65 dB). Stimulus artifact was reduced by alternating the polarity of the acoustic waveform between segments of speech, as described previously [37].

**EEG recording.** Since we were interested in collecting responses which span the auditory pathway as much as possible, we used an eardrum electrode to guarantee a high SNR response from the auditory nerve would be present. Recruitment and limited availability of the clinician to place the electrode prevented us from using it on all subjects, so it was instead used on only a subset of half. The recordings from this electrode for two subjects were excluded from analysis due to poor signal. The electrode, comprising a very small coil of Ag/AgCl wire, was made in-lab following the methods described by Simpson and colleagues [40], with the addition of a chlorination step prior to the final shaping of the electrode. The electrode was placed at the start of the experiment. First, a clear pathway to the eardrum was visually confirmed using a digital otoscope. Then, 2 mL of saline was used to flush the ear canal, and the rest of the EEG electrodes were placed prior to placing the eardrum electrode. The eardrum electrode was then plugged in to a shielded cable (with the outer conductor connected to ground) and input to an EP-preamp (BrainVision, LLC, Greenboro, SC, USA) as the noninverting electrode, with the ipsilateral earlobe used as the inverting electrode. Placement close to the umbo was facilitated using the digital otoendoscope and smooth alligator forceps. Contact with the eardrum was confirmed visually and the subject typically reported audible contact when the electrode contacted the eardrum. Five minutes of clicks at 70, 80, and 90 dB peSPL at 20 stim/s with random timing were presented, and the responses were used to confirm a signal from the eardrum electrode was present. Tympanometry was completed before the experiment to confirm an intact eardrum and aerated middle ear space. Tympanometry was also completed after the experiment to again verify an intact eardrum.

Passive Ag/AgCl electrodes were used to record ABR responses from FCz (in the standard 10–20 montage) referenced to the right and left earlobe with ground on forehead. The earlobes and forehead were prepared prior to electrode placement by scrubbing them with an alcohol swab and NuPrep. The FCz electrode was plugged into a Y-connector to act as the noninverting electrode and connected to two EP-preamps, while the earlobes were used as the inverting electrodes. The preamps were plugged into a splitter which was connected to a BrainVision ActiCHamp or ActiCHamp Plus EEG system (BrainVision, LLC, Greenboro, SC).

A standard 32-channel montage following the 10–20 system and using BrainVision's actiCAP system and active electrodes was used to record cortical responses. For the active electrodes, we aimed for impedances of 10 kΩ or less while for passive electrodes we aimed for impedances of 5 kΩ or less. No impedance target was set for the eardrum electrode as it seemed largely unrelated to SNR and little could be done to adjust it, so we relied on the presence of a response to clicks to confirm good signal quality. EEG was then recorded from all electrodes simultaneously for the duration of the experiment at a sampling rate of 25 kHz (this high sampling rate was used to allow investigation of a possible cochlear microphonic, which was not observed at the moderate stimulus levels we used—10 kHz is sufficient for measuring neural responses).

**Response calculation.** For subcortical responses, the raw response waveform was first downsampled to 10 kHz, then filtered from 30 to 2,000 Hz using a causal first-order Butterworth bandpass filter. For cortical responses, the raw response waveform was causally downsampled to 1,000 Hz by applying an IIR anti-aliasing filter (Chebyshev II filter with a passband edge of 375 Hz, stopband edge of 500 Hz, maximum loss in passband of 1 dB, minimum attenuation in stopband of 60 dB; an additional second-order Butterworth filter with a cutoff of 1,000 Hz was also applied to reduce aliasing of very high-frequency noise) then taking every 25th sample. The data were then referenced to TP9 and TP10, and filtered from 1 to 20 Hz using a causal first-order Butterworth bandpass filter. The EEG was then epoched and corrected for clock drift by resampling, using triggers stamped at the beginning and end of a trial to determine the true sampling rate. Although silent pauses in the stimuli had been truncated to be 500 ms or less during stimulus construction, strong onset and offset responses from the remaining gaps in the speech streams were present in the recordings and were affected by attention. The event-related potentials which show these offset/onset responses, calculated by averaging the EEG at the start of each silence, are shown in S7 Fig. These responses were smeared during our analysis and appeared as artifactual, slow, response components that spanned time lags before and after zero (S8 Fig). To prevent

this component from impacting our analysis, we zeroed the EEG during any point when either audio stream had a silence greater than 490 ms. A Bayesian weighting method was used to improve SNR where each epoch was assigned a weight equal to the inverse of its variance, and all weights were normalized to sum to one for each electrode. Responses were determined through deconvolution as shown in below eq (1):

$$ h = \mathcal{F}^{-1} \left\{ \frac{\sum_N b_n X_n^* Y_n}{\frac{1}{N} \sum_N X_n^* X_n} \right\} \quad (1) $$

(1)

where $h$ denotes the response, $X_n$ the Fast Fourier transform (FFT) of the input stimulus feature for trial $n$ (i.e., the regressor), $Y_n$ the FFT of the output for trial $n$ (EEG data), $b_n$ the weight for trial $n$, $N$ the number of trials, and $\mathcal{F}^{-1}$ the inverse Fourier transform operator. We used the auditory nerve model (ANM) regressor as described previously [38] to determine the responses in conjunction with our peaky speech stimuli [37] for improved SNR [39]. To generate the ANM regressor, the stimuli for each narrator were upsampled to 100 kHz and separately input to a computational model of the auditory nerve [74,75]. We simulated responses from high spontaneous rate fibers at 43 characteristic frequencies from 125 to 16,000 Hz in 1/6th-octave steps, and the stimuli were presented to the model at 65 dB SPL in units of Pascals. The resulting auditory nerve firing rates were downsampled to 10 kHz to match the EEG sampling rate, then summed across characteristic frequency to produce continuous estimates of auditory nerve activity for each stimulus. The modeled response has a latency of 2.75 ms, which has the effect of shifting the TRF earlier. To compensate, we shifted the TRF 2.75 ms to the right after calculation, as in our previous work [38]. The peak value and latency of the CAP and ABR wave V were determined by applying a zero-phase low-pass filter with a cutoff of 500 Hz to each subjects' average response waveform and selecting the minimum value in the 1–4.5 ms time window (CAP) or the maximum value in the 4–9 ms time window (wave V).

Prior to other analysis of the active electrodes, independent component analysis (ICA) was used to remove blinks and eye movement artifacts from each subject's recording. The passive electrodes were not included in ICA analysis since they are only used to examine subcortical responses which are computed with a higher high-pass filter cutoff. To reduce computation time, ICA was computed on 20 min of data, selected from the middle of the experiment. The data were downsampled to 64 Hz and filtered from 1 to 8 Hz using one-pass, zero-phase, noncausal bandpass filters (finite impulse response filter (FIR) length of 211 samples) prior to fitting the ICA solution. Fifteen components were fit to the data and visually inspected for artifacts. Blinks and eye movements were identified by examining the components in the time domain and scalp topographies of each component and selected to be excluded. Typically, 2–3 independent components were removed.

We additionally computed the complex cross-correlation metric described in Forte and colleagues [7] by modifying our analysis pipeline. The EEG was filtered from 100 to 300 Hz using zero-phase bandpass filters. We used the same filters to determine the fundamental waveform, as in previous work [6]. We then took the Hilbert transform of this waveform before computing the cross-correlation of the analytic function with the EEG to generate the complex cross-correlation response waveform and extracting its amplitude envelope. We did not use the Bayesian weighting scheme described above since each cross-correlation is normalized per-trial. This analysis was only carried out on the passive electrodes, as it is intended to isolate responses from the subcortex.

**Statistics.** Statistical analysis was conducted in Python using SciPy and MNE [76,77]. For the subcortical responses, two-tailed paired $t$ tests were performed on selected regions of interest (0–7 ms for the CAP and 0–15 s for the ABR), and the resulting $p$-values were FDR corrected at 0.05. BF were calculated in JASP [78] using the recommended default prior distributions (Cauchy with scale = 0.707) to test if peak sizes were larger or latencies were different for responses to attended versus unattended stimuli. We used paired one-tailed $t$ tests to test if the attended condition produced an enhanced response when compared to the unattended condition (i.e., more negative for the CAP and more positive for ABR wave V). This hypothesis was formed based on cortical response characteristics, where response components are

enhanced for attended stimuli [2,3]. While two-tailed tests are typically preferred for frequentist statistics, one-tailed tests are considered more 'fair' for Bayesian methods [44]. We maintained the same alternative hypothesis for both frequentist and Bayesian tests on the picked peaks for consistency, although the sample-by-sample tests performed over the indicated time windows (Figs 1 and 3) were two-tailed since the direction of the waveform changes over time. Analyses on latencies were two-tailed, since there is no clear prior for the direction of a latency difference had there been one.

Statistical testing of cortical responses was conducted on the 0–300 ms time window using paired, nonparametric, spatiotemporal clustering, with a threshold corresponding to $p = 0.05$. Ten thousand permutations were computed, and any time point where an electrode in the cluster had a $p$-value less than 0.05 was marked as significant.

The maximum values of the envelope complex cross-correlation metric [7] in the 0–20 ms time window (shown in Fig 3) were compared across all subjects first through a binomial test, where the probability of success was set equal to 0.5 and a "success" was defined as when the attended condition produced a larger response than the unattended condition. Additionally, we calculated the ratio of the response size to the attended stimuli and the unattended stimuli for each subject, then tested if the ratio was significantly greater than one using a one-tailed $t$ test, as in Forte and colleagues [7]. Although the ratios are not normally distributed about one, we did not log the ratios before performing the $t$ tests in order to match the analysis in Forte and colleagues. In this instance, the conclusion is unaffected by this detail, as both methods provide a $p$-value greater than the significance threshold.

### Experiment 2: Dichotic stimuli

**Subjects.** Data was recorded from 24 subjects (7 male, 17 female) with a mean age and standard deviation of $28 \pm 6.9$ years (range 19–45). All subjects had normal hearing (defined as audiometric threshold ≤ 20 dB HL at octave frequencies from 500 to 8,000 Hz) which was confirmed by pure-tone audiometry. All subjects reported English as their first language except for two, who had a different first language but had been speaking English daily for over 20 years. Subjects reported no neurological disorders or abnormalities. Written informed consent was obtained to all subjects prior to the experiment and subjects were compensated at an hourly rate. Protocols were approved by the University of Washington Institutional Review Board (No. 39485). All procedures were in accordance with the Declaration of Helsinki.

**Experimental design.** The experimental design for this experiment was the same as the diotic experiment except different sounds were presented to each ear. Because of this, subjects were also given a directional cue (in addition to the speaker sex and story name) to instruct them which story should be attended on each trial. Subjects also responded to three questions instead of two.

**Stimuli.** Stimuli were constructed using the same audiobooks as the diotic experiment. Here, the original (nonpeaky) audio was used, and sounds were presented dichotically. The female speech was presented at its natural pitch, rather than being lowered. Additionally, stimuli were resampled to 24,414 Hz to match the native rate of the real-time processor used to present the stimuli (RP2.1, Tucker Davis Technologies, Alachua, FL, USA). To better drive the ABR, the stimuli were gently high-pass filtered with a first-order Butterworth filter with a cutoff frequency of 1,000 Hz and a slope of 6 dB/octave. The speech stimuli were still intelligible and natural sounding. The stimuli were again split into 64 s epochs with 1 s cosine fade-in and fade-out, and the last 4 s of each trial was repeated as the first 4 s of the next trial and presented ordered so subjects could more easily follow along. Sixty trials of stimuli were presented, resulting in a total of 64 min of stimuli presented to each subject. One subject completed only 44 trials, a total 48 min of stimuli.

**EEG recording.** EEG was recorded using passive Ag/AgCl electrodes, as in the diotic experiment. However, only one differential preamplifier was used. The noninverting electrode was again placed at FCz in the standard 10–20 coordinate system, and the inverting electrode was placed on the left earlobe, with ground at Fpz. Data was recorded at a sampling rate of 10,000 Hz with an online high-pass filter of 0.1 Hz.

**Analysis.** Response calculation and statistical analysis were the same as described for the diotic experiment where the ANM regressor was used to determine the ABR, with an additional step to remove stimulus artifact, which was present

in this experiment since the raw audio was used with no inversion to cancel the artifact. This was accomplished using methods described previously [45]. When determining the subcortical responses, the average stimulus artifact kernel was determined through deconvolution of the EEG recording with the raw stimulus audio. The artifact kernel was then convolved with the stimuli presented on each epoch to estimate the continuous stimulus artifact for each epoch. This estimate was then subtracted from the EEG signal prior to deconvolution. Cortical responses were determined following the same methods used in the diotic experiment.

### Experiment 3: Passive listening to stimuli from Forte and colleagues [7]

**Subjects.** Fourteen subjects (3 male, 10 female, and 1 who did not wish to identify) participated in the passive experiment with a mean age and standard deviation of $23 \pm 5.6$ years (range 18–39). All subjects met the criteria described under the diotic experiment and were compensated in the same way. Written informed consent was obtained from all subjects prior to the experiment and subjects were compensated at an hourly rate. Protocols were approved by the University of Rochester Research Subjects Review Board (No. 1227). All procedures were in accordance with the Declaration of Helsinki.

**Experimental design.** On each trial, subjects were presented with two audiobooks simultaneously, one read by a male narrator and the other by a female narrator. No attentional cues were given, and subjects were instructed to ignore all auditory stimuli in this experiment. To prevent subjects from becoming engaged by the stimuli, subjects were given access to a computer connected to the internet and allowed to distract themselves by watching silent, captioned videos of their choosing or by reading a book they brought with them. Additionally, trial order was randomized so subjects could not follow along with the stories.

**Stimuli.** Subjects were presented with the same stimuli used in Forte and colleagues at the same intensity (76 dB SPL combined). In total, four audiobooks were used in two pairings, matching Forte and colleagues (male speech: *Tales of Troy: Ulysses the Sacker of Cities* and *The Green Forest Fairy Book* narrated by James K. White, female speech: *The Children of Odin* and *The Adventures of Odysseus and the Tale of Troy* narrated by Elizabeth Klett; "target" speech: *Tales of Troy: Ulysses the Sacker of Cities* and *The Adventures of Odysseus and the Tale of Troy*, "distractor" speech: *The Children of Odin* and *The Green Forest Fairy Book*). Stimuli varied in length and were zero-padded for equal trial length, totaling just over 90 min of audio presented to subjects over 32 trials. Stimuli were repeated four times in a row, with the polarity inverted on each trial, to increase SNR and reduce stimulus artifact without affecting the response [7]. We additionally added shielding to the earphone transducers and used shielded electrode cables to further reduce stimulus artifact.

**Response calculation.** Responses were calculated following the method described in Forte and colleagues. The EEG was bandpass filtered between 100 and 300 Hz using zero-phase FIR filters, and data from the first 10 s and after 130 s were discarded, leaving 64 min of data for analysis. The complex fundamental waveform, computed previously through empirical mode decomposition [7], was aligned and cross-correlated with the EEG, as was its Hilbert transform, to determine the complex cross-correlation responses. The maximal amplitude of the envelope of these complex cross-correlation waveforms in the time window 0–20 ms was then taken for comparison across conditions. No envelope was maximal at the edge of these windows, and the latencies were generally close to those described in Forte and colleagues. The responses to male and female speech were then examined separately by comparing the response size for the "target" stimuli with the "distractor" stimuli. The ratio of the responses to the "target" and "distractor" stimuli was calculated for each subject for responses to both male and female speech and used to determine the average ratios.

**Statistics.** If there are no differences in the passive encoding of these stimuli, half of the responses should be larger for the "target" condition than the "distractor" condition (due to random chance). We therefore used a binomial test with the probability of success set to 0.5, where a success is when the response to the "target" stimuli is greater than the response to the "distractor" stimuli. We additionally examined if the ratios of the responses to the "target" and "distractor" stimuli

were significantly greater than unity. This was determined separately for responses to the male and female speech, due to size differences in the responses that are dependent on the narrator gender. Since the distribution of ratios about unity is nonnormal, it would be appropriate to take the log of the ratios prior to performing $t$ tests. However, to match the prior analysis, we did not do a log transform before performing the $t$ tests. In this case, performing the log transform resulted in a still-significant difference for the responses to female speech ($p = 0.001$), but the male response was no longer under the threshold ($p = 0.098$). The overall binomial test was unaffected since it only depended on the direction of the difference for each subject.

## Supporting information

**S1 Fig. Compound action potentials (CAPs) to diotic speech for individual subjects.** The CAPs to attended (red) and unattended (black) speech are shown for all subjects for whom we obtained a response with the eardrum electrode. Grand averages shown in Fig 1a.
(TIF)

**S2 Fig. Auditory brainstem responses (ABRs) to diotic speech for all subjects.** The ABRs to attended (red) and unattended (black) speech are shown for all subjects. Grand averages shown in Fig 1b.
(TIF)

**S3 Fig. Cortical sound encoding is affected by selective attention in a dichotic listening task.** Cortical responses to attended, **a**, and unattended, **b**, stimuli. Each electrode is represented by a different color trace (key in upper left). Significant differences were observed in the 91–110, 125–218, and 242–300 ms regions (shaded on the time-series plots), as determined through paired, two-tailed spatiotemporal clustering methods ($p < 0.05$ for at least one electrode in the interval). Scalp topographies are shown for selected time points of 100 and 170 ms, denoted by a vertical dashed line on the time series plots. **c**, The difference waveform and scalp topographies at the selected time points. Asterisks indicate the electrodes that were significantly different across conditions at that time.
(TIF)

**S4 Fig. Individual responses for the diotic experiment calculated using the complex cross-correlation method [7].** Responses to the attended (red) and unattended (black) speech are shown. Response peaks are shown in Fig 3.
(TIF)

**S5 Fig: Individual responses for the passive listening experiment calculated using the complex cross-correlation method [7].** Responses to the "target" (red) and "distractor" (black) speech are shown. The "target" response can be clearly seen to be larger than the "distractor" response in several subjects, despite the task being passive. Response peaks are shown in Fig 4.
(TIF)

**S6 Fig. Two control analyses.** In order to ensure counterbalancing, all stimuli were presented exactly twice—once attended, and once unattended (order counterbalanced). Neither responses to attended, **a**, nor unattended, **c**, stimuli are impacted by whether the stimulus was novel or repeated. To address the possibility of the 2-hour experiment time impacting results (e.g., through waning interest), we analyzed separately the first 30, **b**, and last 30 min, **d**. There was no effect of attention at any time during the experiment.
(TIF)

**S7 Fig. Offset responses during continuous speech differ across attention conditions.** Cortical event-related potentials to the offsets in the **a**, attended and **b**, unattended speech streams. Offsets of speech stimuli occurred at zero on the time axis, and onset of the following speech occurred within the 490–500 ms time window. Each electrode

is represented by a different color trace (key in upper left). Significant differences were determined through paired, two-tailed spatiotemporal clustering methods ($p < 0.05$ for at least one electrode in the shaded region). Scalp topographies are shown for selected time points of 100 and 670 ms, denoted by a vertical dashed line on the time series plots. **c**, The difference waveform and scalp topographies. Asterisks indicate the electrodes that were significantly different at the selected time point.

(TIF)

**S8 Fig. Artifactual acausal response shifts resulting from large responses to sound offsets.** Without eliminating attention-dependent offset responses during the silent pauses (Fig 6), a spurious shift in the TRFs resulted. The shift was slow and began well before 0 ms latency. It also depended on the specific TRF regressor used, and while present with the ANM regressor, **a**, it was even more pronounced when regressing against glottal pulses [39], **b**. Not eliminating the shift could have led to complications when interpreting the results.

(TIF)

## Acknowledgments

We thank Yathida Melody Anankul, Ashlynne G. Xavier, Tiffany Waddington, and Susan A. McLaughlin for assistance with data collection. We are grateful to Tobias Reichenbach and colleagues for freely sharing experimental data and stimuli, and to Skyler G. Jennings for teaching us how to make and use the eardrum electrodes.

## Author contributions

**Conceptualization:** Thomas J. Stoll, Adrian K. C. Lee, Ross K. Maddox.

**Data curation:** Thomas J. Stoll, Ross K. Maddox.

**Formal analysis:** Thomas J. Stoll, Ross K. Maddox.

**Funding acquisition:** Ross K. Maddox.

**Investigation:** Thomas J. Stoll, Nathan D. Vandjelovic, Melissa J. Polonenko, Ross K. Maddox.

**Methodology:** Thomas J. Stoll, Nathan D. Vandjelovic, Melissa J. Polonenko, Nadja R. S. Li, Adrian K. C. Lee, Ross K. Maddox.

**Project administration:** Ross K. Maddox.

**Software:** Ross K. Maddox.

**Supervision:** Adrian K. C. Lee, Ross K. Maddox.

**Validation:** Thomas J. Stoll, Ross K. Maddox.

**Visualization:** Thomas J. Stoll, Ross K. Maddox.

**Writing – original draft:** Thomas J. Stoll, Ross K. Maddox.

**Writing – review & editing:** Thomas J. Stoll, Ross K. Maddox.

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
