## [Editor Report · Decision Letter 0]

27 Jan 2025

Dear Dr Maddox, 

Thank you for submitting your manuscript entitled "The when and where of auditory selective attention" for consideration as a Research Article by PLOS Biology.

Your manuscript has now been evaluated by the PLOS Biology editorial staff as well as by an academic editor with relevant expertise and I am writing to let you know that we would like to send your submission out for external peer review as a SHORT REPORT.

Once your full submission is complete, your paper will undergo a series of checks in preparation for peer review. After your manuscript has passed the checks it will be sent out for review. To provide the metadata for your submission, please Login to Editorial Manager (https://www.editorialmanager.com/pbiology) within two working days, i.e. by Jan 29 2025 11:59PM.

Kind regards,

Christian

Christian Schnell, PhD

Senior Editor

PLOS Biology

cschnell@plos.org

---

## [Decision Letter · Decision Letter 1]

12 Mar 2025

Dear Dr Maddox,

Thank you for your patience while your manuscript "The when and where of auditory selective attention" was peer-reviewed at PLOS Biology. It has now been evaluated by the PLOS Biology editors, an Academic Editor with relevant expertise, and by several independent reviewers. 

In light of the reviews, which you will find at the end of this email, we would like to invite you to revise the work to thoroughly address the reviewers' reports.

As you will see below, the reviewers think that the study your is overall well executed and provides important insights. However, in addition to the comments regarding the lack of methodological details and required textual revisions, for example regarding the length of the experiment and the impact of stimulus repetition, there is also a control experiment that will be required to address the reviewers' concerns, mentioned by Reviewer 2.

Reading the cross-comments, R2 agrees mostly with R1 and reiterates that the requested controls are necessary to support the claims in his opinion.

Given the extent of revision needed, we cannot make a decision about publication until we have seen the revised manuscript and your response to the reviewers' comments. Your revised manuscript is likely to be sent for further evaluation by all or a subset of the reviewers.

**IMPORTANT - SUBMITTING YOUR REVISION**

*Re-submission Checklist*

*Published Peer Review*

*PLOS Data Policy*

*Blot and Gel Data Policy*

Sincerely,

Christian

Christian Schnell, PhD

Senior Editor

PLOS Biology

cschnell@plos.org

REVIEWS:

Reviewer #1: The manuscript by Stoll et al. is an attempt to address conflicting results about whether or not subcortical auditory responses are modulated by selective attention to speech. Data were recorded in three experiments, where two concurrent audiobooks were presented to listeners, and they either had to pay attention to one of them (Experiments 1&2) or ignore them and engage in another task of their choice (Experiment 3). They utilize a unique recording setup to obtain neural responses from the auditory nerve (CAP), brainstem (ABR), and cortex (traditional scalp-EEG), which allowed them to separate the different levels of response.

Counter to results reported in previous studies, and primarily by Forte et al., they authors find no evidence for modulation of subcortical responses by selective attention, but replicate the well-established findings of cortical modulations. Moreover, they use the stimuli from the Forte et al. study in a passive-listening paradigm to claim that previous results may be related to stimulus-specific effect rather than selective attention.

I found this paper to be quite interesting and well executed, and I appreciate the methodological rigor for demonstrating the reliability of a "null" result. I believe that this work makes an important contribution in furthering this debate, and - while it may not be conclusive - provides compelling and important empirical evidence against selective-attention effects at the brainstem level (at least when listening to 2-hour long speech!).

I do, however, have several questions and missing details that I would appreciate if the authors could clarify:

1. The stimulus-presentation scheme is not clear to me. Authors write that "The stories were ordered such that subjects could follow along, with each stimulus played twice (once attended, once ignored) so they would not miss sections of the story. At the end of each trial, subjects were asked two comprehension questions to keep them engaged in the task".

But many details are missing: Did they switch their attention between the male/female storied on each trial? Were the repetitions consecutive or were all the stimuli repeated in the 2nd half of the experiment?

2. And on a more methodological level - is it possible that repeating the same content twice could affect speech tracking? One could imagine that hearing a story that I had previously listened to might be more distracting than an entirely new story.. were any control-analysis conducted to ensure this did not have an effect? 

>> the experiment was extremely long (over 2 hours!). I would expect attention to deteriorate over time or just general effects of fatigue. Can you comment on this? Did the behavioral performance change over time? This seems critical to me, since selective-attention is instructed here, but listeners may or may not have complied with this instruction, especially in such a long experiment. So. verifying that they were indeed paying attention as required is critical for interpreting the lack of selective-attention effects at the subcortical level.

3. Experiment 3: While I greatly appreciate the rationale of using the stimuli used by Forte et al. in a passive-listening design, in order to assess whether previously reported effects are potentially due to stimulus-specific features. While the distribution of single-subject data is nice to see, I am not fully convinced of this alternative explanation for several reasons: 

a) Results of passive-listening paradigms are very difficult to interpret, especially if they are long, since it's hard to know what participants are actually doing/paying attention to. It is quite possible that for at least some of the time, they listened to the speech, and that one stimulus 'grabbed' their attention more consistently. Do the authors have any insight into what participants actually did during this time? It seems unlikely to me that they were actively engaged in another task 100% of the time… 

b) I wonder why the authors did not use these materials in a selective-attention paradigm as well - if they were able to show that similar responses are obtained during passive listening and "active/selective" listening, then perhaps this might be more convincing.

c) I also wonder why the authors chose not to show the waveforms of the neural responses obtained in the passive-listening experiment, which offer more direct comparisons to Experiments 1&2 and to prior work. 

d) The authors claim that previous results may be due to "unbalanced" presentation of target and distractor speech (i.e., that they were not repeated), however I wonder if they have a hypothesis as to what speech-features might be driving the preference for one speech-stimulus over the other in their passive-listening conditions. This would be important for testing this possibility or controlling for acoustic differences in future studies (especially since repeating the same stimuli also has substantial drawbacks, as noted above). 

4. For completion, it would be helpful if the authors presented the full results for Experiment 2 as well (CAP, ABR and scalp-EEG results). 

5. The "control" analysis for offset responses described briefly in the methods and in Extended Data Fig 1 sounds potentially important, but was not entirely clear to me. Could the authors please elaborate on this? One thing I found particularly confusing is that silences are not expected to be synchronized across attended and unattended speech, nor do they have constant durations (of 490-500ms). So, how should we interpret the ERPs shown here? 

Also, it would be nice what "artifact" the authors were trying to correct for in the TRF by zeroing the EEG during silences. Perhaps other studies suffer from a similar "problem"? 

Reviewer #2: This study investigates whether attention modulates subcortical responses using EEG and an eardrum electrode. While the methodology is rigorous, the framing of the findings is too general. For example, the title and the framing of the paper suggest there is no attentional modulation in subcortical areas, but this conclusion may depend on the type of stimulus used, the recording technique, and behavioral demands. These should be made more clear by making the claims more specific, than what is said in the title currently. 

Prior studies (e.g., Slee & David, 2015) have reported subcortical attentional effects with simpler stimuli. One possibility is that attentional modulation emerges at different stages depending on stimulus complexity—when auditory features are easily separable, modulation could occur subcortically, but in complex scenes like multitalker environments, cortical processing may be required first. The study does not establish "when and where" attention acts, but rather what is observable with EEG. This distinction needs to be clearer.

The paper would also benefit from a stronger connection to theoretical models of brain function. If subcortical attention effects are absent, does this support a strictly feedforward model of auditory selection? What does this imply about how the brain processes complex acoustic scenes? 

A critical missing control is in Experiment 3, where the authors argue that prior studies mistakenly attributed acoustic differences to attentional effects. However, their passive listening design does not confirm that subjects were truly disengaged. Without behavioral verification, it remains possible that subjects were involuntarily engaged with one stream over another, possibly with the more salient/engaging story. A stronger test would be an internal control within their own dataset: splitting attended and ignored stimuli into non-overlapping time segments (e.g., first half attended, second half ignored). If their claim is correct, this artificial separation should create a spurious "attention effect" purely due to acoustic differences. Without this test, their critique of prior work remains speculative. Conducting this additional control would provide a stronger argument that prior findings were driven by acoustic confounds rather than genuine attentional modulation.

Minor points:

- Justification of dichotic stimulus presentation requires more citations. (e.g. this sentence: . If the sounds were spatially separated, a much simpler scheme could be employed wherein the better ear was favored)

- In Figure 2: Would a difference plot (attend minus unattended) or an alternative visualization help clarify the effects, particularly for early vs. late responses?

Reviewer #3: This study examines how selective attention influences sound encoding along the auditory pathway. Using a novel experimental setup, Stoll et al. simultaneously record neural responses from multiple levels of the auditory system. While prior research suggested that attention might modulate subcortical processing, their findings provide strong evidence that selective attention does not impact sound encoding in the auditory periphery or brainstem, with attentional effects first emerging in the cortex.

The study is well-executed, incorporating various control experiments (e.g., diotic vs. dichotic conditions, data reanalysis using methods from Forte et al.) to account for potential null findings. Additionally, the manuscript offers a plausible explanation for at least one previous report of attentional modulation in subcortical processing.

I find this to be an excellent piece of work that merits publication. However, I have one consideration: the authors chose to repeat each stimulus twice, once attended and once unattended conditions. They have a good reason for making this choice, as changing stimuli between conditions introduces acoustic differences that might confound attention effects (as the authors suggest happened in the study by Forte et al.). However, the impact of stimulus repetition should not be overlooked—not only in terms of changes in neural responses to first vs second stimulus presentation, but also regarding cognitive factors such as task engagement and cognitive load. Could the repetition of stimuli have reduced the likelihood of detecting an attentional effect?

---

## [Decision Letter · Decision Letter 2]

14 Aug 2025

Dear Ross,

Thank you for your patience while we considered your revised manuscript "The auditory brainstem response to natural speech is not affected by selective attention" for publication as a Short Reports at PLOS Biology. This revised version of your manuscript has been evaluated by the PLOS Biology editors and two of the original reviewers.

Based on the reviews, we are likely to accept this manuscript for publication, provided you satisfactorily address the following data and other policy-related requests:

* Short Reports can only have four figures. We suggest combining figures 3/4 and 5/6 into one figure each to not exceed the limit. Figure 6 could also be moved to the supplementary information, in which case you'd need to combine only figures 3 and 4 into one.

* Please add the links to the funding agencies in the Financial Disclosure statement in the manuscript details.

* Please include the approval/license number and IRB names for all three experiments. 

* Please specify for all three experiments whether the participants provided written or verbal consent, and include information whether the study has been conducted according to the principles expressed in the Declaration of Helsinki.

* DATA POLICY:

Regardless of the method selected, please ensure that you provide the individual numerical values that underlie the summary data displayed in the following figure panels as they are essential for readers to assess your analysis and to reproduce it: 1CD

* CODE POLICY

We expect to receive your revised manuscript within two weeks. 

*Published Peer Review History*

*Press*

Sincerely,

Christian

Christian Schnell, PhD

Senior Editor

cschnell@plos.org

PLOS Biology

Reviewer remarks:

Reviewer #1: The authors have addressed my comments successfully, and I appreciate their candid discussion of the limitations of the study alongside its added benefit and unique contribution. 

Reviewer #3: I thank the authors for replying to my question. I have no further comments.

---

## [Editor Report · Decision Letter 3]

3 Sep 2025

Dear Ross,

Thank you for your patience while we considered your revised manuscript "The auditory brainstem response to natural speech is not affected by selective attention" for publication as a Short Reports at PLOS Biology. This revised version of your manuscript has been evaluated by the PLOS Biology editors and the Academic Editor.

While most of the editorial requests have been addressed in the revision, a few points remain open:

* DATA POLICY:

Regardless of the method selected, please ensure that you provide the individual numerical values that underlie the summary data displayed in the following figure panels as they are essential for readers to assess your analysis and to reproduce it: 1CD

* CODE POLICY

* I noted that you provide a link to a repository, but this was not accessible. In case the code is deposited there with a DOI, that would be sufficient. But please ensure the repository is accessible so we can ensure it meets the requirements. 

We expect to receive your revised manuscript within two weeks. 

*Published Peer Review History*

*Press*

Sincerely,

Christian

Christian Schnell, PhD

Senior Editor

cschnell@plos.org

PLOS Biology

---

## [Editor Report · Decision Letter 4]

9 Sep 2025

Dear Ross,

Thank you for the submission of your revised Short Reports "The auditory brainstem response to natural speech is not affected by selective attention" for publication in PLOS Biology. On behalf of my colleagues and the Academic Editor, David Poeppel, I am pleased to say that we can in principle accept your manuscript for publication, provided you address any remaining formatting and reporting issues. These will be detailed in an email you should receive within 2-3 business days from our colleagues in the journal operations team; no action is required from you until then. Please note that we will not be able to formally accept your manuscript and schedule it for publication until you have completed any requested changes.

While you attend to those requests to come, please also make sure to reference the source data. Please cite the location of the data clearly in the legend of Figure 1, e.g. “The data underlying this Figure can be found in doi:10.18112/openneuro.ds006434 Code/Results/exp1Diotic/mags+lats.csv”

PRESS

Sincerely, 

Christian

Christian Schnell, PhD

Senior Editor

PLOS Biology

cschnell@plos.org